# Efficiently applying attention to sequential data with the Recurrent Discounted Attention unit

## ABSTRACT

Recurrent Neural Networks architectures excel at processing sequences by modelling dependencies over different timescales. The recently introduced *Recurrent Weighted Average* (RWA) unit captures long term dependencies far better than an LSTM on several challenging tasks. The RWA achieves this by applying attention to each input and computing a weighted average over the full history of its computations. Unfortunately, the RWA cannot change the attention it has assigned to previous timesteps, and so struggles with carrying out consecutive tasks or tasks with changing requirements. We present the *Recurrent Discounted Attention* (RDA) unit that builds on the RWA by additionally allowing the discounting of the past.

We empirically compare our model to RWA, LSTM and GRU units on several challenging tasks. On tasks with a single output the RWA, RDA and GRU units learn much quicker than the LSTM and with better performance. On the multiple sequence copy task our RDA unit learns the task three times as quickly as the LSTM or GRU units while the RWA fails to learn at all. On the Wikipedia character prediction task the LSTM performs best but it followed closely by our RDA unit. Overall our RDA unit performs well and is sample efficient on a large variety of sequence tasks.

## 1 INTRODUCTION

Many types of information such as language, music and video can be represented as sequential data. Sequential data often contains related information separated by many timesteps, for instance a poem may start and end with the same line, a scenario which we call long term dependencies. Long term dependencies are difficult to model as we must retain information from the whole sequence and this increases the complexity of the model.

A class of model capable of capturing long term dependencies are Recurrent Neural Networks (RNNs). A specific RNN architecture, known as Long Short-Term Memory (LSTM) (Hochreiter and Schmidhuber, 1997), is the benchmark against which other RNNs are compared. LSTMs have been shown to learn many difficult sequential tasks effectively. They store information from the past within a hidden state that is combined with the latest input at each timestep. This hidden state can carry information right from the beginning of the input sequence, which allows long term dependencies to be captured. However, the hidden state tends to focus on the more recent past and while this mostly works well, in tasks requiring equal weighting between old and new information LSTMs can fail to learn.

A technique for accessing information from anywhere in the input sequence is known as attention. The attention mechanism was introduced to RNNs by Bahdanau et al. (2014) for neural machine translation. The text to translate is first encoded by a bidirectional-RNN producing a new sequence of encoded state. Different locations within the encoded state are focused on by multiplying each of them by an attention matrix and calculating the weighted average. This attention is calculated for each translated word. Computing the attention matrix for each encoded state and translated word combination provides a great deal of flexibility in choosing where in the sequence to attend to, but

the cost of computing these matrices grows as a square of the number of words to translate. This cost limits this method to short sequences, typically only single sentences are processed at a time.

The *Recurrent Weighted Average* (RWA) unit, recently introduced by Ostmeyer and Cowell (2017), can apply attention to sequences of any length. It does this by only computing the attention for each input once and computing the weighted average by maintaining a running average up to the current timestep. Their experiments show that the RWA performs very well on tasks where information is needed from any point in the input sequence. Unfortunately, as it cannot change the attention it assigns to previous timesteps, it performs poorly when asked to carry out multiple tasks within the same sequence, or when asked to predict the next character in a sample of text, a task in which new information is more important than old.

We introduce the *Recurrent Discounted Attention* (RDA) unit, which extends the RWA by allowing it to discount the attention applied to previous timesteps. As this adjustment is applied to all previous timesteps at once, it continues to be efficient. It performs very well both at tasks requiring equal weighting over all information seen and at tasks in which new information is more important than old.

The main contributions of this paper are as follows:

1. We analyse the Recurrent Weighted Average unit and show that it cannot output certain simple sequences.

2. We propose the Recurrent Discounted Attention unit that extends the Recurrent Weighted Average by allowing it to discount the past.

3. We run extensive experiments on the RWA, RDA, LSTM and GRU units and show that the RWA, RDA and GRU units are well suited to tasks with a single output, the RDA performs best on the multiple sequence copy task while the LSTM unit performs better on the Hutter Prize Wikipedia dataset.

Our paper is setout as follows: we present the analysis of the RWA (sections 3 and 4) and propose the RDA (section 5). The experimental results (section 6), discussion (section 7) and conclusion follow (section 8).

## 2    RELATED WORK

Recently many people have worked on using RNNs to predict the next character in a corpus of text. Sutskever et al. (2011) first attempted this on the Hutter Prize Wikipedia datasets using the MRNN archtecture. Since then many architectures (Graves, 2013; Chung et al., 2015; Kalchbrenner et al., 2015; Rocki, 2016; Zilly et al., 2016; Ha et al., 2016; Chung et al., 2016) and regularization techniques (Ba et al., 2016; Krueger et al., 2016) have achieved impressive performance on this task, coming close to the bit-per-character limits bespoke compression algorithms have attained.

Many of the above architectures are very complex, and so the Gated Recurrent Unit (GRU) is a much simpler design that achieves similar performance to the LSTM. Our experiments confirm previous literature (Chung et al., 2014) that reports it performing very well.

Attention mechanisms have been used in neural machine translation by Bahdanau et al. (2014). Xu et al. (2015) experimented with hard-attention on image where a single location is selected from a multinomial distribution. Xu et al. (2015) introduced the global and local attention to refer to attention applied to the whole input and hard attention applied to a local subset of the input.

An idea related to attention is the notion of providing additional computation time for difficult inputs. Graves (2016) introduce shows that this yields insight into the distribution of information in the input data itself.

Several RNN architectures have attempted to deal with long term dependencies by storing information in an external memory (Graves et al., 2014; 2016).

## 3    RECURRENT WEIGHTED AVERAGE

At each timestep the Recurrent Weighted Average model uses its current hidden state $h_{t-1}$ and the input $x_t$ to calculate two quantities:

1. The features $z_t$ of the current input:

$$u_t = W_u \cdot x_t + b_u$$
$$g_t = W_g \cdot [x_t, h_{t-1}] + b_g$$
$$z_t = u_t \odot \tanh g_t$$

where $u_t$ is an unbounded vector dependent only on the input $x_t$, and $\tanh g_t$ is a bounded vector dependent on the input $x_t$ and the hidden state $h_{t-1}$.

*Notation: $W$ are weights, $b$ are biases, $(\cdot)$ is matrix multiplication, and $\odot$ is the element-wise product.*

2. The attention $a_t$ to pay to the features $z_t$:

$$a_t = e^{W_a \cdot [x_t, h_{t-1}] + b_a}$$

The hidden state $h_t$ is then the average of of the features $z_t$, *weighted* by the attention $a_t$, and squashed through the hyperbolic tangent function:

$$h_t = \tanh \left( \frac{\sum_{i=1}^{t} z_i \odot a_i}{\sum_{i=1}^{t} a_i} \right)$$

This is implemented efficiently as a running average:

$$n_t = n_{t-1} + z_t \odot a_t$$
$$d_t = d_{t-1} + a_t$$
$$h_t = \tanh \left( \frac{n_t}{d_t} \right)$$

where $n_t$ is the numerator and $d_t$ is the denominator of the average.

## 4    PROPERTIES OF THE RECURRENT WEIGHTED AVERAGE

The RWA shows superior experimental results compared to the LSTM on the following tasks:

1. Classifying whether a sequence is longer than 500 elements.
2. Memorising short random sequences of symbols and recalling them at any point in the subsequent 1000 timesteps.
3. Adding two numbers spaced randomly apart on a sequence of length 1000.
4. Classifying MNIST images pixel by pixel.

All of these tasks require combining the full sequence of inputs into a single output. It makes perfect sense that an average over all timesteps would perform well in these tasks.

On the other hand, we can imagine tasks where an average over all timesteps would not work effectively:

1. Copying many input sequences from input to output. It will need to forget sequences once they have been output.
2. Predicting the next character in a body of text. Typically, the next character depends much more on recent characters than on those from the beginning of the text.
3. Outputting the parity of an input sequence $h_t = -1^t c$   for   $0 < c < 1$.

All of these follow from the property that $d_t$ is monotonically increasing in $t$, which can be seen from $a_t > 0$ and $d_t = d_{t-1} + a_t$. As $d_t$ becomes larger, the magnitude of $a_t$ must increase to change the value of $h_t$. This means that it becomes harder and harder to change the value of $h_t$ to the point where it almost becomes fixed. In the specific case of outputting the sequence $h_t = -1^t c$ we can show that $a_t$ must grow geometrically with time.

**Lemma 1** *Let the task be to output the sequence $h_t = -1^t c$ for $0 < c < 1$. Let $h_t$ be defined by the equations of the Recurrent Weighted Average, and let $z_t$ be bounded and $f_h$ be a continuous, monotonically increasing surjection from $\mathbb{R} \to (-1, 1)$.*
*Then, $a_t$ grows geometrically with increasing $t$.*

*Proof. Provided in Appendix A.* □

**Corollary 2** *If $a_t$ is also bounded then it cannot grow geometrically for all time and so the RWA cannot output the sequence $h_t = -1^t c$.*

Corollary 2 suggests that the Recurrent Weighted Average may not actually be Turing Complete.

Overall, these properties suggest the the RWA is a good choice for tasks with a single result, but not for sequences with multiple results or tasks that require forgetting.

## 5 THE RECURRENT DISCOUNTED ATTENTION UNIT

The RDA uses its current hidden state $h_{t-1}$ and the input $x_t$ to calculate three quantities:

1. The features $z_t$ of the current input are calculated identically to the RWA:

$$u_t = W_u \cdot x_t + b_u$$
$$g_t = W_g \cdot [x_t, h_{t-1}] + b_g$$
$$z_t = u_t \odot \tanh g_t$$

2. The attention $a_t$ to pay to the features: $z_t$

$$a_t = f_a(W_a \cdot [x_t, h_{t-1}] + b_a)$$

Here we generalize attention to allow any function $f_a$ which is non-negative and monotonically increasing. If we choose $f_a = \exp$, then we recover the RWA attention function.

3. The discount factor $\gamma_t$ to apply to the previous values in the average

$$\gamma_t = \sigma(W_\gamma \cdot [x_t, h_{t-1}] + b_\gamma)$$

where $\sigma$ is the sigmoid/logistic function defined as $\sigma(x) = \frac{1}{1+e^{-x}}$.

We use these values to calculate a discounted moving average. This discounting mechanism is crucial in remediating the RWA's inability to forget the past

$$n_t = n_{t-1} \odot \gamma_t + z_t \odot a_t$$
$$d_t = d_{t-1} \odot \gamma_t + a_t$$
$$h_t = f_h\left(\frac{n_t}{d_t}\right)$$

Here we generalize RWA further by allowing $f_h$ to be any function, and we also introduce a final transformation to the hidden state $h_t$ to produce the output

$$o_t = f_o(h_t)$$

### 5.1 CHOICES FOR THE ATTENTION FUNCTION $f_a$, HIDDEN STATE FUNCTION $f_h$ AND OUTPUT FUNCTION $f_o$

The attention function $f_a(x)$ is a non-negative monotonically increasing function of $x$. There are several possible choices:

- $f_a(x) = e^x$ - This is used in the RWA.
- $f_a(x) = \max(0, x)$ - Using a ReLU allows the complete ignoring of some timesteps with linearly increasing attention for others.
- $f_a(x) = \ln(1 + e^x)$ - The softplus function is a smooth approximation to the ReLU.
- $f_a(x) = \sigma(x)$ - Using the sigmoid limits the maximum attention an input can be given.

The domain of the hidden activation function $f_h$ is the average $\frac{n_t}{d_t}$. This average is bounded by the minimum and maximum values of $z_t$. Possible choices of $f_h$ include:

- $f_h(\frac{n_t}{d_t}) = \tanh(\frac{n_t}{d_t})$ - This is used in the RWA. We observed that the range of $\frac{n_t}{d_t}$ mostly remained in the linear domain of $\tanh$ centred around 0, suggesting that using this was unneccessary.
- $f_h(\frac{n_t}{d_t}) = \frac{n_t}{d_t}$ - The identity is our choice for $f_h$ in the RDA.

Possible choices for the output function $f_o$ are:

- $f_o(h_t) = h_t$ - The RWA uses the identity as its hidden state has already been transformed by $\tanh$.
- $f_o(h_t) = \tanh(h_t)$ - The output can be squashed between $[-1, 1]$ using $\tanh$.

## 6 EXPERIMENTS

We ran experiments to investigate the following questions:

1. Which form of the RDA works best? (Section 6.2)
2. The RWA unit works remarkably well for sequences with a single task. Does the RDA unit retain this strength? (Section 6.3)
3. We expect the RWA unit to struggle with consecutive independent tasks. Does this happen in practice and does the RDA solve this problem? (Section 6.4)
4. How does the RDA unit scale up to very long sequences? We test character prediction on the Hutter Prize Wikipedia dataset. (Section 6.5)
5. How does the RDA unit compare to RWA, LSTM and GRU units? Are some units more suited to certain types of tasks than others? (Section 7)

We provide plots of the training process in Appendix B.

### 6.1 IMPLEMENTATION DETAILS

For all tasks except the Wikipedia character prediction task, we use 250 recurrent units. Weights are initialized using Xavier initialization (Glorot and Bengio, 2010) and biases are initialized to 0, except for forget gates and discount gates which are initialized to 1 (Gers, Schmidhuber, and Cummins, 2000). We use mini-batches of 100 examples and backpropagate over the full sequence length. We train the models using Adam Kingma and Ba (2014) with a learning rate of 0.001. Gradients are clipped between -1 and 1.

For the Wikipedia task, we use a character embedding of 64 dimensions, followed by a single layer of 1800 recurrent units, and a final softmax layer for the character prediction. We apply truncated backpropagation every 250 timesteps, and use the last hidden state of the sequence as the initial hidden state of the next sequence to approximate full backpropagation.

All of our experiments are implemented in TensorFlow (Abadi et al., 2016).

### 6.2 EMPIRICAL EVALUATION OF RDA ACTIVATION FUNCTIONS

We ran our experiments with different combinations of $f_a$ and $f_o$ and found the following:

| Addition | |
|---|---|
| **Model** | Steps until loss $< 0.001$ |
| GRU | 2036 |
| LSTM | $> 10000$ |
| RDA-exp-tanh | 1781 |
| RDA-sigmoid-id | 2016 |
| RWA | 1735 |

Table 1: Addition: steps until loss $< 0.001$.

| Classify Length | |
|---|---|
| **Model** | Steps until accuracy = 1.0. |
| GRU | 71 |
| LSTM | 776 |
| RDA-exp-tanh | 164 |
| RDA-sigmoid-id | 414 |
| RWA | 133 |

Table 2: Classify: steps until accuracy = 1.0.

| MNIST | |
|---|---|
| **Model** | Test Set Accuracy |
| GRU | 0.985 |
| LSTM | 0.114 |
| RDA-exp-tanh | 0.985 |
| RDA-sigmoid-id | 0.987 |
| RWA | 0.979 |

Table 3: MNIST test set accuracy.

| MNIST permuted | |
|---|---|
| **Model** | Permuted Test Set Accuracy |
| GRU | 0.944 |
| LSTM | 0.915 |
| RDA-exp-tanh | 0.905 |
| RDA-sigmoid-id | 0.913 |
| RWA | 0.899 |

Table 4: MNIST permuted test set accuracy.

- Using a ReLU for the attention function $f_a$ almost always fails to train. Using a Softplus for $f_a$ is much more stable than a ReLU. However, it doesn't perform as well as using sigmoid or exponential attention.

- Exponential attention performs well in all tasks, and works best with the $\tanh$ output function $f_o(h_t) = \tanh(h_t)$. We refer to this as *RDA-exp-tanh*.

- Sigmoid attention performs well in all tasks, and works best with the identity output function $f_o(h_t) = h_t$. We refer to this as *RDA-sigmoid-id*.

- It is difficult to choose between RDA-exp-tanh and RDA-sigmoid-id. RDA-exp-tanh often trains faster, but it sometimes diverges with NaN errors during training. RDA-sigmoid-id trains slower but is more stable, and tends to have better loss.
  We include results for both of them.

### 6.3 SINGLE TASK SEQUENCES

Here we investigate whether sequences with a single task can be performed as well with the RDA as with the RWA.

Each of the four tasks detailed below require the RNN to save some or all of the input sequence before outputting a single result many steps later.

1. *Addition* - The input consists of two sequences. The first is a sequence of numbers each uniformly sampled from $[0, 1]$, and the second consists of all zeros except for two ones which indicate the two numbers of the first sequence to be added together. (Table 1)

2. *Classify length* - A sequence of length between 1 and 1000 is input. The goal is to classify whether the sequence is longer than 500.
   All RNN architectures could learn their initial hidden state for this task, which improved performance for all of them. (Table 2)

3. *MNIST* - The task is supervised classification of MNIST digits. We flatten the 28x28 pixel arrays into a single 784 element sequence and use RNNs to predict the digit class labels. This task challenges networks' ability to learn long-range dependencies, as crucial pixels are present at the beginning, middle and end of the sequence. We implement two variants of this task:

   (a) Sequential - the pixels are fed in from the top left to the bottom right of the image. (Table 3)

| Copy | | | Multicopy | | |
|---|---|---|---|---|---|
| **Model** | Steps until accuracy $> 0.999$ | | **Model** | Steps until accuracy $> 0.99$ | |
| GRU | 5329 | | GRU | 3984 | |
| LSTM | $> 20000$ | | LSTM | 4048 | |
| RDA-exp-tanh | 11831 | | RDA-exp-tanh | 1114 | |
| RDA-sigmoid-id | 9840 | | RDA-sigmoid-id | 1316 | |
| RWA | 5660 | | RWA | $> 10000$ | |

Table 5: Copy: steps until accuracy $> 0.999$    Table 6: Multicopy: steps until accuracy $> 0.99$

| Hutter Prize Wikipedia | |
|---|---|
| **Model** | BPC |
| Stacked LSTM (Graves, 2013) | 1.67 |
| MRNN (Sutskever et al., 2011) | 1.60 |
| GF-LSTM (Chung et al., 2015) | 1.58 |
| Grid-LSTM (Kalchbrenner et al., 2015) | 1.47 |
| MI-LSTM (Wu et al., 2016) | 1.44 |
| Recurrent Memory Array Structures (Rocki, 2016a) | 1.40 |
| HyperNetworks (Ha et al., 2016) | 1.35 |
| LayerNorm HyperNetworks (Ha et al., 2016) | 1.34 |
| Recurrent Highway Networks (Zilly et al., 2016) | 1.32 |
| LayerNorm LSTM† | 1.39 |
| HM-LSTM | 1.34 |
| LayerNorm HM-LSTM | 1.32 |
| GRU (our implementation) | 1.535 |
| LSTM (our implementation) | 1.492 |
| RDA-exp-tanh | N/A |
| RDA-sigmoid-id | 1.529 |
| RWA | 5.067 |
| PAQ8hp12 (Mahoney, 2005) | 1.32 |
| decomp8 (Mahoney, 2009) | **1.28** |

Table 7: Bits per character on the Hutter Prize Wikipedia test set

  (b) Permuted - the pixels of the image are randomly permuted before the image is fed in.
      The same permutation is applied to all images. (Table 4)

 4. *Copy* - The input sequence starts with randomly sampled symbols. The rest of the input is
    blanks except for a single recall symbol. The goal is to memorize the starting symbols and
    output them when prompted by the recall symbol. All other output symbols must be blank.
    (Table 5)

## 6.4    MULTIPLE SEQUENCE COPY TASK

Here we investigate whether the different RNN units can cope with doing the same task repeatedly.

The tasks consists of multiple copying tasks all within the same sequence. Instead of having the recall symbol randomly placed over the whole sequence it always appears a couple of steps after the sequence being memorized. This gives room for 50 consecutive copying tasks in a length 1000 input sequence. (Table 6)

## 6.5    WIKIPEDIA CHARACTER PREDICTION TASK

The standard test for RNN models is character-level language modelling. We evaluate our models on the Hutter Prize Wikipedia dataset enwik8, which contains 100M characters of 205 different symbols including XML markup and special characters. We split the data into the first 90M characters for the training set, the next 5M for validation, and the final 5M for the test set. (Table 7)

# 7 DISCUSSION

We start our discussion by describing the performance of each individual unit.

Our analysis of the RWA unit showed that it should only work well on the single task sequences and we confirm this experimentally. It learns the single sequence tasks quickly but is unable to learn the multiple sequence copy task and Wikipedia character prediction task.

Our experiments show that the RDA unit is a consistent performer on all types of tasks. As expected, it learns single task sequences slower than the RWA but it actually achieves better generalization on the MNIST test sets. We speculate that the cause of this improvement is because the ability to forget effectively allows it to compress the information it has previously processed, or perhaps discounting the past should be considered as changing the attention on the past and the RDA is able to vary its attention on previous inputs based on later inputs. On the multiple sequence copy task the RDA unit was far superior to all other units learning three times as fast as the LSTM and GRU units. On the Wikipedia character prediction task the RDA unit performed respectably, achieving a better compression rate than the GRU but worse than the LSTM.

The LSTM unit learns the single task sequences slower than all the other units and often fails to learn at all. This is surprising as it is often used on these tasks as a baseline against which other archtectures are compared. On the multiple sequence copy task it learns slowly compared to the RDA units but solves the task. The Wikipedia character prediction task is where it performs best, learning much faster and achieving better compression than the other units.

The GRU unit works very well on single task sequences often learning the fastest and achieving excellent generalization on the MNIST test sets. On the multiple sequence copy task it has equal performance to the LSTM. On the Wikipedia character prediction task it performs worse than the LSTM and RDA units but still achieves a good performance.

We now look at how our results show that different neural network architectures are suited for different tasks.

For our single output tasks the RWA, RDA and GRU units work best. Thus for similar real work applications such as encoding a molecule into a latent representation, classification of genomic sequences, answering questions or language translation, these units should be considered before LSTM units. However, our results are yet to be verified in these domains.

For sequences that contain an unknown number of independent tasks the RDA unit should be used.

For the Wikipedia character prediction task the LSTM performs best. Therefore we can't recommend RWA, RDA or GRU units on this or similar tasks.

# 8 CONCLUSION

We analysed the Recurrent Weighted Average (RWA) unit and identified its weakness as the inability to forget the past. By adding this ability to forget the past we arrived at the Recurrent Discounted Attention (RDA). We implemented several varieties of the RDA and compared them to the RWA, LSTM and GRU units on several different tasks. We showed that in almost all cases the RDA should be used in preference to the RWA and is a flexible RNN unit that can perform well on all types of tasks.

We also determined which types of tasks were more suited to each different RNN unit. For tasks involving a single output the RWA, RDA and GRU units performed best, for the multiple sequence copy task the RDA performed best, while on the Wikipedia character prediction task the LSTM unit performed best. We recommend taking these results into account when choosing a unit for real world applications.

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

# Appendices

## A  MATHEMATICAL PROOFS

**Lemma 1** *Let the task be to output the sequence $h_t = -1^t c$ for $0 < c < 1$. Let $h_t$ be defined by the equations of the Recurrent Weighted Average, and let $z_t$ be bounded and $f_h$ be a continuous, monotonically increasing surjection from $\mathbb{R} \to (-1, 1)$.*
*Then, $a_t$ grows geometrically with increasing $t$.*

*Proof*

Given that the activation $f_h$ is a continuous, monotonically increasing surjection from $\mathbb{R}$ to $(-1, 1)$, we know that there are two values $x_+$ and $x_-$ such that $f(x_+) = c$ and $f(x_-) = -c$. Define $x_+ - x_- = \delta$.

Then for every even integer $i$, we have $\frac{n_i}{d_i} = x_+$ and $\frac{n_{i+1}}{d_{i+1}} = x_-$.

From the definitions of $n_t$ and $d_t$ we have

$$\frac{n_{i+1}}{d_{i+1}} = \frac{n_i + z_{i+1} a_{i+1}}{d_i + a_{i+1}} = x_-$$

Substituting $n_i = d_i x_+$ and rearranging yields

$$a_{i+1} = \frac{d_i (x_- - x_+)}{z_{i+1} - x_-} \geq \frac{d_i |\delta|}{|z|_{max} + |x|_{max}}$$

where $|z|_{max} = \max\{|z|\}$ and $|x|_{max} = \max\{|x_+|, |x_-|\}$.

Substituting this into $d_{i+1} = d_i + a_{i+1}$ gives us

$$d_{i+1} \geq d_i \left( 1 + \frac{|\delta|}{|z|_{max} + |x|_{max}} \right)$$

By a similar argument, we have the same growth for odd integers $d_{i+2} \geq d_{i+1} \left( 1 + \frac{|\delta|}{|z|_{max} + |x|_{max}} \right)$ and we get geometric growth of $d_t$.

From the definition of $d_t$ we have

$$a_i = d_{i+1} - d_i \geq d_i \frac{|\delta|}{|z|_{max} + |x|_{max}}$$

As $d_t$ grows geometrically, then so does $a_t$. $\square$

**Corollary 2** *If $a_t$ is also bounded then it cannot grow geometrically for all time and so the RWA cannot output the sequence $h_t = -1^t c$*

*Proof* Assume the RWA can output the sequence $h_t = -1^t c$. As $a_t$ grows geometrically, it is unbounded, but this is a contradiction. $\square$

# B ILLUSTRATED EMPIRICAL RESULTS

We include figures of the loss function learning curves during training. In the case of MNIST, we report on validation accuracy instead. These experiments provide evidence that the two flavours of the RDA unit consistently perform close to the best across a broad range of tasks. Figures are best viewed in colour.

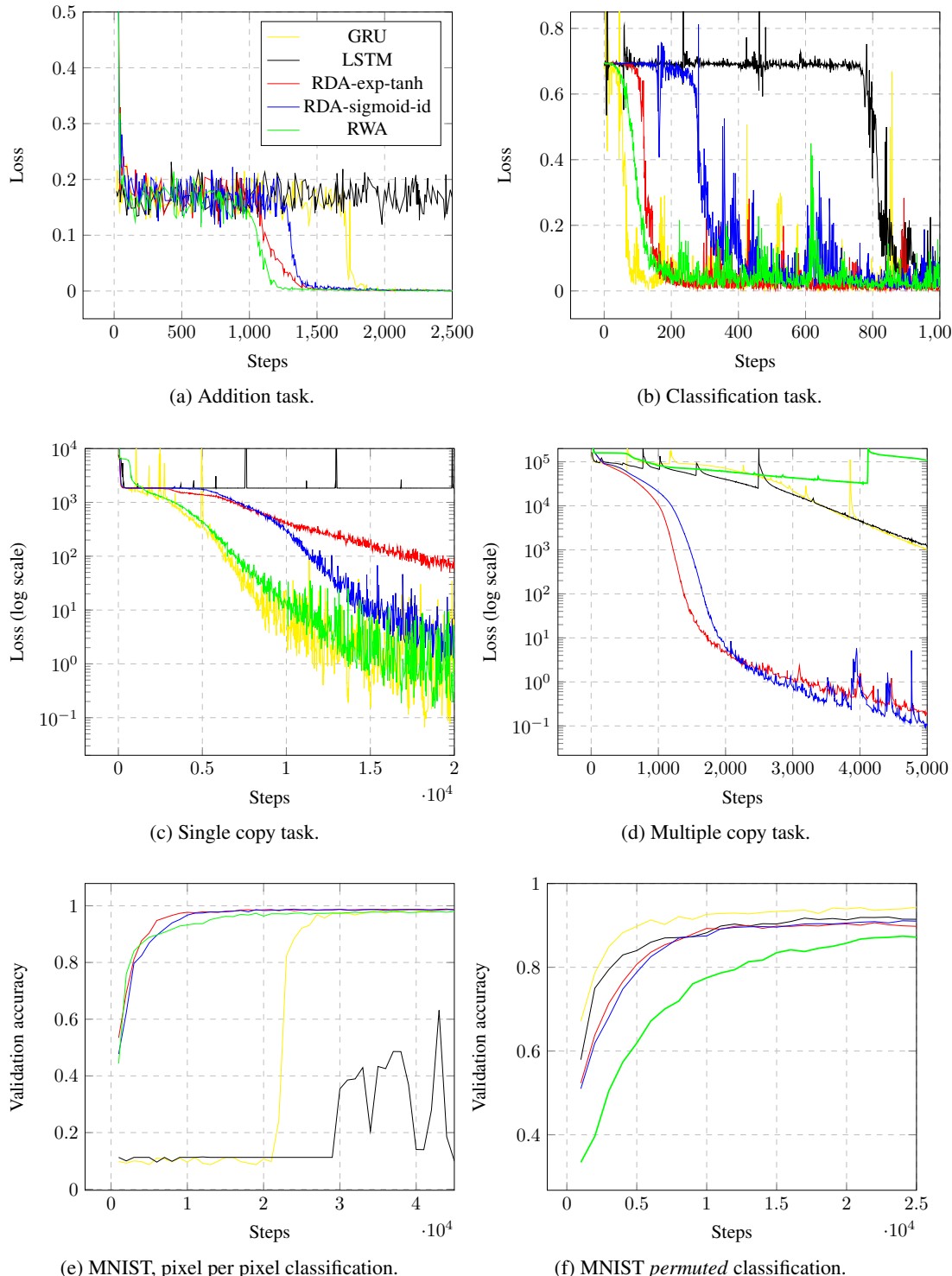

(a) Addition task.

(b) Classification task.

(c) Single copy task.

(d) Multiple copy task.

(e) MNIST, pixel per pixel classification.

(f) MNIST *permuted* classification.

