# OpenReview forum: "Efficiently applying attention to sequential data with the Recurrent Discounted Attention unit"
_ICLR.cc/2018/Conference — Reject_

### Official Review · AnonReviewer3 · 2017-11-26
**Improvement upon a rarely used technique still not showing great results yet.**

**Rating:** 4
**Confidence:** 5

**Review:**

The authors present RDA, the Recurrent Discounted Attention unit, that improves upon RWA, the earlier introduced Recurrent Weighted Average unit, by adding a discount factor. While the RWA was an interesting idea with bad results (far worse than the standard GRU or LSTM with standard attention except for hand-picked tasks), the RDA brings it more on-par with the standard methods.

On the positive side, the paper is clearly written and adding discount to RWA, while a small change, is original. On the negative side, in almost all tasks the RDA is on par or worse than the standard GRU - except for MultiCopy where it trains faster, but not to better results and it looks like the difference is between few and very-few training steps anyway. The most interesting result is language modeling on Hutter Prize Wikipedia, where RDA very significantly improves upon RWA - but again, only matches a standard GRU or LSTM. So the results are not strongly convincing, and the paper lacks any mention of newer work on attention. This year strong improvements over state-of-the-art have been achieved using attention for translation ("Attention is All You Need") and image classification (e.g., Non-local Neural Networks, but also others in ImageNet competition). To make the evaluation convincing enough for acceptance, RDA should be combined with those models and evaluated more competitively on multiple widely-studied tasks.

---

### Official Review · AnonReviewer2 · 2017-11-27

**Rating:** 6
**Confidence:** 4

**Review:**

This paper extends the recurrent weight average (RWA, Ostmeyer and Cowell, 2017) in order to overcome the limitation of the original method while maintaining its advantage. The motivation of the paper and the approach taken by the authors are sensible, such as adding discounting was applied to introduce forget mechanism to the RWA and manipulating the attention and squash functions.

The proposed method is using Elman nets as the base RNN. I think the same method can be applied to GRUs or LSTMs. Some parameters might be redundant, however, assuming that this kind of attention mechanism is helpful for learning long-term dependencies and can be computed efficiently, it would be nice to see the outcomes of this combination.

Is there any explanation why LSTMs perform so badly compared to GRUs, the RWA and the RDA?
Overall, the proposed method seems to be very useful for the RWA.

---

### Official Review · AnonReviewer1 · 2017-11-28
**An Extension to RWA**

**Rating:** 3
**Confidence:** 4

**Review:**

Summary:
This paper proposes an extension to the RWA model by introducing the discount gates to computed discounted averages instead of the undiscounted attention. The problem with the RWA is that the averaging mechanism can be numerically unstable due to the accumulation operations when computing d_t.

Pros:
- Addresses an issue of RWAs.

Cons:
-The paper addresses a problem with an issue with RWAs. But it is not clear to me why would that be an important contribution.
-The writing needs more work.
-The experiments are lacking and the results are not good enough.

General Comments:

This paper addresses an issue regarding to RWA which is not really widely adopted and well-known architecture, because it seems to have some have some issues that this paper is trying to address. I would still like to have a better justification on why should we care about RWA and fixing that model.

The writing of this paper seriously needs more work.  The Lemma 1 doesn't make sense to me, I think it has a typo in it, it should have been (-1)^t c instead of -1^t c.

The experiments are only on toyish and small scale tasks. According to the results the model doesn't really do better than a simple LSTM or GRU.

---

### Decision · Program_Chairs · 2018-01-29
**ICLR 2018 Conference Acceptance Decision**

**Decision:**

Reject

**Comment:**

RDA improves on RWA, but even so, the model is inferior to the other standard RNN models. As a result R1 and R3 question the motivation for the use of this model -- something the authors should motivate.